# Fin Erosion of *Salmo salar* (Linnaeus 1758) Infested with the Parasite *Caligus rogercresseyi* (Boxshall & Bravo 2000)

**DOI:** 10.3390/ani10071166

**Published:** 2020-07-09

**Authors:** Margarita P. González, Sandra L. Marín, Melinka Mancilla, Hernán Cañon-Jones, Luis Vargas-Chacoff

**Affiliations:** 1Programa de Doctorado en Ciencias de la Acuicultura, Escuela de Graduados, Universidad Austral de Chile, Los Pinos s/n, Balneario Pelluco, Puerto Montt 5500000, Chile; 2Departamento de Salud Hidrobiológica, División de Acuicultura, Instituto de Fomento Pesquero, Balmaceda 252, Puerto Montt 5501248, Chile; 3Instituto de Acuicultura, Universidad Austral de Chile, Los Pinos s/n, Balneario Pelluco, Puerto Montt 5500000, Chile; smarin@spm.uach.cl; 4Laboratorio de Interacciones Ecológicas (LINTEC), Universidad Austral de Chile, Los Pinos s/n, Balneario Pelluco, Puerto Montt 5500000, Chile; melinka.mancilla@uach.cl or; 5Facultad de Medicina Veterinaria y Agronomía, Universidad de las Américas, Avda. Manuel Montt 948, Santiago 7500972, Providencia, Chile; hcanon@udla.cl; 6Instituto de Ciencias Marinas y Limnológicas, Universidad Austral de Chile, Independencia 641, Valdivia 5110566, Chile; luis.vargas@uach.cl; 7Centro Fondap de Investigación de Altas Latitudes (IDEAL), Universidad Austral de Chile, Valdivia 5110566, Chile

**Keywords:** *Caligus rogercresseyi*, experimental infestation, fin damage, *Salmo salar*, sea lice, welfare

## Abstract

**Simple Summary:**

Sea lice can generate wounds on fish and they are associated with the consumption of mucus, skin, and blood, inducing extensive epithelial erosion and inflammation. Knowledge related to the fin condition of fish with sea lice infections is scarce, limiting the range of parameters regarding their welfare status of fish. In the present study, the fin condition and two stress parameters were assessed in *Caligus rogercresseyi* infected fish. An association between fin erosion, increasing fish stress and parasite abundances was observed, suggesting that *C. rogercresseyi* infection is a possible cause of fin degeneration in Atlantic salmon.

**Abstract:**

Fin condition is a simple indicator of fish welfare, which anticipates detrimental effects on fish in aquaculture systems. This study evaluated the fin condition of *Salmo salar* at different abundances of the parasite *Caligus rogercresseyi*. Fish were exposed to infestation with copepodids and the cohort was allowed to develop to the adult stage. The relative fin index was measured. Significant differences between infested and control fish for both pectoral and anal fins were observed. Moreover, there were significant negative relationships between fin condition and parasite abundances for pectoral, anal, and pelvic fins, suggesting that infestations with *C. rogercresseyi* could be a possible cause for fin damage in Atlantic salmon. Moreover, this damage was associated with increased stress levels, suggesting that damage can be related to physiological changes on infested fish. According to these results, pectoral fin assessments have the potential to provide information on the welfare of fish with *C. rogercresseyi* infestation. Determining the causes of poor fin development may improve fish welfare, even when infested by parasites.

## 1. Introduction

Salmon louse are ectoparasitic copepods that infest wild and salmonid farmed fish [1], causing significant economic and social impacts in countries where salmon and trout industries exist [2,3,4]. In Chile, *Caligus rogercresseyi* is the most important ectoparasite affecting the salmon industry [5,6], and has a wide host range spanning from nonsalmonid native fish such as *Eleginops maclovinus* to salmonids species [7]. Their life cycle includes eight developmental stages separated by successive molting. The free-living phases are nauplius I, II, and the infestive planktonic copepodid, while the attached parasitic stages are chalimus I to IV and, male and female adult stages [5].

Damage from sea lice on fish is associated with its feeding activity, which includes consuming mucus, skin, and blood. This cause surface damage to fish resulting in mucus breakdown, which leads to open sores and lesions [8] that increase the risk of secondary infections [1]. Effects of sea lice on fish physiology have been described for *Lepeophtheirus salmonis, Caligus elongatus*, and *C. rogercresseyi. L. salmonis* induced high plasma cortisol and glucose levels [9,10], a decrease in plasma proteins in *Salmo salar* [11], and high plasma levels of lactate, chloride, and osmolality in *Salmo trutta* [12]. *Caligus elongatus* can cause erosion of dermal tissue principally in the head region [13], parasite that attaches mainly to dorsal and ventral body zones of fish and fins [14]. For *C. rogercresseyi*, González et al. [6] described that an infestation with more than six adult parasites per fish induced high plasma cortisol and glucose levels, low plasma proteins levels and an increase in the number of small skin mucous cells. Information regarding external effects such as fin erosion on fish induced by *C. rogercresseyi* infestation is scarce.

There is an increasing interest in quality attributes such as fish appearance, fish should look healthy, aesthetically pleasing, and without visible signs of suffering or malformation [15]. Further, for commercial farming, fish welfare is considered a key factor in many decisions related to husbandry practices and long term production planning [16]. Regarding sea lice, parasites are strictly controlled and regulated by achieving low lice levels, therefore, frequent handling and treatment associated with delousing may be a more serious welfare issue than the lice themselves [16]. Following these requirements, some fish variables have been proposed as indicators of fish welfare, e.g., fish physiology, health, and behavior [17]. Increased levels of cortisol is a physiological indicator of stress [18]. However, measuring cortisol is time-consuming and is not practical for routine monitoring due to the handling required to extract fish blood [19]. Ellis, et al. [20] described a noninvasive assay for measuring free cortisol from rainbow trout in the water, although it does not allow individual measurement.

Huntingford et al. [21] reported several indicators that indicate decreased fish welfare e.g., increasing ventilation rate, changes in fish behavior, reduced food intake, presence of abnormalities, and injuries such as fin damage. Fin erosion or damage has been defined as “an injury to live tissue including nerve endings” on fins [22], and it was successfully demonstrated as an external welfare indicator in salmonids [19,23]. Hoyle, et al. [24] mentioned that fin condition analysis is a direct method that eliminates the killing of the fish, has no special equipment requirements, and is a noninvasive method. Fin damage is a rapid and simple indicator, and with complementary indices, it may constitute a barometer of fish welfare; anticipating detrimental effects that can compromise the fishes’ economic value [15]. To date, no studies have evaluated fin condition in fish subjected to *C. rogercresseyi* infestations.

Even though physiological changes in fish caused by sea lice are good indicators of fish welfare, they are difficult to implement for routine monitoring on fish farms, and as mentioned by Huntingford and Kadri [25] routine monitoring of welfare requires systems that are effective, economical, and practical for use under farming conditions. Bosakowski and Wagner [26] reported a positive linear relationship between total (standard) body and fin lengths, constituting the relative fin index as an indicator of fin condition. Any disruption of these relationships could be explained by negative impacts produced by an external factor. The present study aimed to evaluate the fin condition of *S. salar* infested with the parasite *C. rogercresseyi*. Determining the relationships between abundance and developmental stages of the parasite and the condition of the fin is relevant: firstly to identify whether there is an association between parasite effects and fin damage, and secondly to gather knowledge that support the establishment of preventive measures for improving fish welfare during infection with this parasite. Further, this characterization will allow fin damage to be evaluated as a potential indicator of the fish condition during *C. rogercresseyi* parasitism.

## 2. Materials and Methods

### 2.1. Animals and Maintenance Conditions

Experiments were carried out at the facilities of Universidad Austral de Chile, located in Puerto Montt, Chile (41°29′32.32″ S–72°53′45.95″ W). Two hundred and eighty *S. salar* smolts of the SNMH12 group without parasites were transported from Rauco farm at Chiloé, Chile to the university facilities. These fish were randomly distributed in 11 circular tanks (300 L) and acclimated for 30 days. The tanks had white walls, with a net on the bottom of the tank to avoid fish escaping through pipes, a plastic flute in the middle of the tank which induced water flow and circular blue airstones. The initial fish weight showed no significant differences among fish from different tanks therefore the mean fish weight grouping of all fish was 85.25 g (n = 280, 95% of confidence interval (CI): 81.01–89.45 g). Fish were maintained in a flow-through system (7.5 L min^−1^) with seawater filtered at 2 µm and a photoperiod of 8:16 h light/dark [7]. Fish were fed ad libitum daily using commercial dry pellets at a ratio of 1.5% body weight and fish satiety was used as an indicator to end fish feeding. Lack of appetite was not observed at any time. The food was sourced from Biomar CPK 100 diet (proteins: 41–49%; lipids: 17–25%; Pargua, Chile). Salinity (mean: 31 psu; 95% of CI: 30.86–1.18 psu), temperature (mean; 11.3 °C; 95% of CI: 11.27–11.37 °C), and oxygen saturation (mean: 86.8%; 95% of CI: 86.42–87.25%) were registered daily using a YSI 85 multifunction meter (Yellow Springs Instruments Inc., Yellow Springs, OH, USA). The experiments complied with guidelines of the Comisión Nacional de Ciencias y Tecnología de Chile (CONICYT), and the Universidad Austral de Chile for the use of laboratory animals, all experiments were authorized by the Committee of Bioethics and Use of Animals in Experiments of Universidad Austral de Chile (certificate N° 148/2014).

### 2.2. Production of Infective Stages and Fish Infestation

Copepodids for fish infestation were obtained from *C. rogercresseyi* females collected from fish maintained at the Laboratory of Ecological Interactions of Universidad Austral de Chile in Puerto Montt, Chile. Females were transported to the laboratory and egg-strings were carefully removed and classified as mature (dark-colored) and immature (white-colored). Only mature eggs were incubated in 5 L aquaria containing seawater at 32 ppm (filtered at 5 µm and UV-disinfected), in a culture chamber kept at 12 °C, with a 12:12 photoperiod, and constant aeration. Infestation was carried out when 90% of the parasites reached the copepodid stage. The infestation trials were performed according to Marín et al. [7], by stopping the water flow for six hours and incorporating the copepodids into the tanks, allowing infestation. The control group was exposed to similar conditions to the infection groups, but without copepodids.

### 2.3. Experimental Design

Fin condition was evaluated by using the experimental design described in González et al. [6]. This design included two experimental and one control group with two replicates each. Infestation pressure of 50 copepodids per fish (50 IP) and 100 copepodids per fish (100 IP) were used to obtain a gradient of parasite abundance on fish, since incremental levels of infestation by *C. rogercresseyi* copepodids does not produce a linear response in terms of settlement on *S. salar* [27] and therefore there is high variability in the infestation success [7]. Control fish groups were not exposed to parasites. Photographs of fish fins and blood samples were obtained at 1-day preinfestation and at 1, 8, 16, and 22 days post infestation (dpi) to find the stages of copepodid, chalimus I–II (Ch I–II), chalimus III–IV (Ch III–IV), and adults (female and male).

### 2.4. Sampling Procedure

On each sampling day, each fish was individually netted and subjected to an anesthetic dose of benzocaine chlorhydrate (30 ppm). After confirming the anesthetic state 3 or 4 described by Close et al. [28], fish blood was collected from the caudal peduncle using 1 mL syringes in less than one minute and added to 1.5 mL heparinized tubes (1000 units of porcine heparin per 1 mL of 0.9% NaCl). Plasma was separated by centrifuging the whole blood (5 min, 2000× *g*, 4 °C) and then stored at −20 °C until analysis. Fish, nets, tray water, and trays were inspected for detached parasites, which were counted and classified by the developmental stage according to González and Carvajal [5] and quantified as the number of parasites per fish. Each fish was identified individually with a number and put into a tray of 5 cm height by 35 cm wide; half-filled with seawater. A Sony DSC-W110 (Tokyo, Japan) camera was used to take digital photographs of the body and rayed fins (dorsal, caudal, both pectoral, and pelvic, and anal fins) were taken on the left and right side of fish. The camera was mounted on a standard device or tripod which allowed taking photographs at a standardized height. Photographs were taken while the fish was still anesthetized. Measurements of the standard length of the body and wet weight were recorded for each fish during sampling.

### 2.5. Fin Damage Quantification

Fin damage was quantified by determining the relative fin index (RFI) as described by Bosakowski and Wagner [26] and Cañon Jones et al. [29]. This index was calculated for each individual by dividing its fin length (to the tip of the longest fin-ray) by the maximum standard length of the body. Lower RFI values indicate shorter fins, and higher RFI values indicate longer fins. These values allowed the fin reduction percentage of infested fish to be calculated and compared to control fish. Photographs were analyzed through ImageJ software and standardized to a known distance.

### 2.6. Plasma Parameters

Plasma glucose was quantified using commercial kits from Spinreact (Glucose-HK Ref. 1001200, Girona, Spain) adapted for 96-well microplates [30]. Plasma cortisol was quantified by ELISA using a commercial kit from DIA Source Immuno Assays S.A. (Cortisol Ref. KAPDB270, Louvain-la-Neuve, Belgium). The interassay coefficient of variation at 50% binding was 5.6% (n = 4), whereas the mean intra-assay coefficient of variation (calculated from the sample duplicates) was 5.6%. The mean percentage of recovery was 95% (n = 4). Main cross-reactivity of 100% given by cortisol was detected with prednisolone (13.6%), deoxycorticosterone (7.2%), cortisone (6.2%), and corticosterone (7.6%) [31,32].

### 2.7. Statistical Analysis

Prior to the analysis, RFI values of paired fins were averaged (pectoral, pelvic, and superior and lower caudal fins), since no significant differences were observed between paired fins. After testing the normality and homogeneity of the variances of the relative fin index (%) for each fin data, plasma glucose and cortisol corroboration of replication for each treatment was performed and then one-way ANOVA analysis between the infested and control fish groups was performed for each fin at each sampling day [33]. The relationship between fin condition and parasitic abundances was determined by considering the number of parasites per fish as a continuous independent variable (gradient of parasites) and using simple regression analysis. Prior to the analysis, assumptions of normality, independence, and constant error variance were checked using Shapiro-Wilk, Durbin-Watson [12], and Cochran C, Hartley, and Bartlett tests in the residuals, respectively. To identify significant regressions with explanations above 40% of variance, r, r^2^, and *p*-value parameters of each regression were estimated. All analyses were performed in STATISTICA v. 10 (StatSoft, Inc., Palo Alto, CA, USA).

## 3. Results

### 3.1. Caligus Rogercresseyi Development and Abundances

Tanks with infested fish showed a prevalence of 100% throughout the experiment. At one day postinfestation (dpi) all parasites were copepodids, and at 8 dpi all were Ch I–II (Table 1). At 16 dpi, 97% of the parasites were Ch III–IV and 3% were adults (2% males, 1% females). At 22 dpi, all parasites had reached the adult stage; 54% were males and 46% females (Table 1). The abundances varied significantly among tanks for each sampling day (Table 2), being 41.3, 29.8, 25.5, 19.8, and 8.5 parasites per fish at 1, 8, 16, and 22 dpi, respectively.

### 3.2. Fin Damage, Plasma Glucose, and Cortisol Levels

Prior to infestation, no significant differences between infested and control fish groups were observed for the relative fin index (Table 3). Dorsal and caudal fins did not show significant differences between the control and infested fish groups at any sampling day.

Pectoral fins showed significant differences between fish groups at 8 dpi (F = 4.64; *p* = 0.04) and 16 dpi (F = 4.61; *p* = 0.04; Figure 1A). For pelvic fins, significant differences were observed at 1 dpi (F = 9.13; *p* = 0.01) and at 22 dpi (F = 4.69; *p* = 0.04; Figure 1B). Anal fins showed significant differences between fish groups only at 8 dpi (F = 4.59; *p* = 0.04; Figure 1C). RFI values of infested fish groups showing significant differences were lower than control fish group values. Plasma glucose showed significant results only at 16 dpi (F = 10.16; *p* = 0.004; Figure 2A), and plasma cortisol only at 22 dpi (F = 8.33, *p* = 0.009; Figure 2B). Significant linear regressions between the RFI (%) and the number of parasites per fish were found for caudal fin at 1 dpi (Figure 3A; y = 19.6 − 0.01x, r = −0.38, r^2^ = 0.14, *p* = 0.04), for pectoral (Figure 3B; y = 16.9 − 0.04x, r = −0.67, r^2^ = 0.44, *p* = 0.00), pelvic (Figure 3C; y = 11.75 − 0.03x, r = −0.50, r^2^ = 0.25, *p* = 0.01), and anal fins (Figure 3D; y = 10.9 − 0.02x, r = −0.49, r^2^ = 0.24, *p* = 0.01) at 8 dpi, for pectoral fin for both at 16 dpi (Figure 3E; y = 16.6 − 0.04x, r = −0.47, r^2^ = 0.22, *p* = 0.01) and at 22 dpi (Figure 3F; y = 17.3 − 0.06x, r = −0.46, r^2^ = 0.22, *p* = 0.02). In all these instances, the RFI (%) decreased with the increment of parasitic abundances. Relationships among physiological parameters and fin condition were estimated. At 16 dpi, RFI (%) values for dorsal fin showed significant linear regressions with both plasma glucose (Figure 4A; y = 19.01 − 1.64x, r = −0.51, r^2^ = 0.26, *p* = 0.01) and cortisol levels (Figure 4B; y = 11.7 − 0.13x, r = −0.45, r^2^ = 0.20, *p* = 0.04). At 22 dpi, mean RFI (%) values of pectoral fins (Figure 4C; y = 19.3 − 0.15x, r = −0.56, r^2^ = 0.31, *p* = 0.01) and pelvic fins (Figure 4D; y = 13.3 − 0.11x, r = −0.51, r^2^ = 0.26, *p* = 0.04) showed a significant linear regressions with plasma cortisol levels. Again, in all these cases, the RFI (%) decreased with the increment of the physiological parameters.

## 4. Discussion

*Caligus rogercresseyi* induced a negative effect on the condition of some fins of *S. salar*. The group of infested fish showed a relative fin index value lower than the control fish group, suggesting that chalimus stages can reduce the length of pectoral, pelvic and anal fins, and the adult stage affecting mainly the pectoral fins. Bosakowski and Wagner [26] found reductions (10% to 50%) in dorsal fin length of rainbow, cutthroat, and brown trout under hatchery conditions when compared to wild fish. Further, Stejskal et al. [34] indicated that the pectoral, second dorsal, ventral, and anal fins of intensively cultured perch showed reductions up to 52%, 49%, 35%, and 28%, respectively. In this study, the pectoral, pelvic, and anal fins were affected by different stages of *C. rogercresseyi*, causing reductions of relative fin index of ~1.1% and ~1.8% during the chalimus stages, and ~0.7% at adult stage infestation, respectively, which means shorter fins for infested fish.

This fin length reduction was also related to increased parasite abundance, suggesting that higher abundances can increase fin damage. The relationship was observed for the caudal fins of infested fish with copepodids individuals, for pectoral, pelvic, and anal fins of fish infested with chalimus I and II individuals, and for pectorals fins of fish infested with chalimus III and IV, and also adults individuals. Studies about the external effects produced by *C. rogercresseyi* are scarce, however, some studies reported that the parasite can negatively affect physiological conditions. Wells et al. [12] estimated threshold values obtained from nonlinear regression among physiological variables and parasite abundances. These authors indicated that 6 and 13, 13 and 24, and 12 parasites per fish altered the glucose, lactate, and osmolality levels, respectively, of *Salmo trutta* infested by *L. salmonis* adults. Moreover, González et al. [6] reported thresholds values of 6–7, 14–15, and 20–21 parasites per fish for plasma cortisol and glucose, skin mucous cells, and for proteins, amino acids, triglycerides, and lactate, respectively, of *S. salar* infested by *C. rogercresseyi* adults.

Many causes for fin damage and body lesions have been described, e.g., water quality, high stocking density, feeding, handling, among others [35]. For *S. salar,* the hatchery environment conditions reduced pectoral and dorsal fins, with severe degeneration in the caudal fin [35]. Further, Turnbull et al. [36] mentioned that the dorsal fin was the most severely damaged, since it is the target of aggressive fish attacks under farm conditions, agreeing with Brockmark et al. [37] who mentioned that high density induced high fish aggression, and consequently, high fin damage, since the latter is a useful indicator of aggressive behavior [38]. To our knowledge, this is the first study that described a direct relationship between the reduction of fish welfare of Atlantic salmon (observed through external effects) and increasing abundances of *C. rogercresseyi*, suggesting that this parasite can be a risk factor, as well as a cause for fin deterioration. However, it was not possible to identify the origin of the damage, since it could be induced directly by the parasite to the live tissue or due to stress aggressive behavior among fish infested by the parasite. Further, it could be induced by other behaviors like increased rubbing against tanks and jumping out of the water. Future studies should evaluate the effects of parasite infestations along with other variables; evaluating the effects of parasites on attachment zones using indicators of aggressive behavior (e.g., bites and splits (separations of tissue between rays) on fins).

Moreover, stressful situations can also erode fish fins. Udomkusonsri et al. [39] described that hybrid striped bass exhibit an extremely rapid erosion on distal edges of fins after 15 min of confinement. Previous studies described that parasites induce a stress response in fish, e.g., a high infestation of *L. salmonis* adults for a short period resulted in increased plasma cortisol levels in *S. trutta* (~20 µg dl^−1^) [12], and *S. salar* at 21 dpi [40], and an increase of 1.75-fold higher than the control group for *Oncorhynchus keta* [41]. González et al. [6] suggested that six or more adult parasites of *C. rogercresseyi* induced physiological stress in *S. salar* (~120 g), reaching 23 ug dl^−1^ in plasma cortisol levels. It is possible that stress induced by advanced stages of *C. rogercresseyi* could also have an impact on fin erosion of dorsal, pectoral, and pelvic fins since the negative relationship between physiological variables and the relative fin index values of these fins were also observed. When fish showed high levels of both glucose and cortisol in plasma (both evidence of stressed fish) shorter fins were also observed, agreeing with the observations of Udomkusonsri et al. [39]. In the present study, it is not possible to separate stress and the effects of parasites, considering both as a mixed effect on fins.

Fins are known as attachment sites for a variety of ectoparasites, and relationships between fin erosion and parasite infestation have been described, e.g., in sea lice [22,42]. Tully et al. [43] mentioned that *L. salmonis* copepodids and chalimus stages attached preferentially on the dorsal fin, and preadults and adults on dorsal and anal fins [42]. Treasurer and Bravo [44] indicated that chalimus stages of *C. rogercresseyi* were distributed across the body of the fish, and the adult stage was concentrated on the abdomen. Skin lesions and fin erosion could be produced by the attachment filament or be due to the oral cone during feeding [42,45,46]. Damage at the attachment and feeding sites on the fins of Chinook (*Oncorhynchus tshawytscha*) and Atlantic salmon (*S. salar*) was described by Johnson and Albright [47] as being characterized by extensive epithelial erosion and inflammation. In the present study, pectoral, pelvic, and anal fins were the most affected by *C. rogercresseyi*, but this trend could not be correlated with an attachment preference (since parasites count per fin was not performed).

Fins are used by fish for several functions, e.g., movement, stability, swimming, among others [48], e.g., dorsal fin acts as a body stabilizer during swimming [35], pectoral and caudal fins as locomotory fins [49], pelvic and anal fins allow hovering, low speed turning, and deceleration of the body [50], and caudal fins are used for burst swimming [35]. Consequently, a detrimental condition on fins may reduce fishes’ swimming ability, and the damage caused by heavy copepod infestation may render fins nonfunctional [50], which would negatively affect fish welfare.

Moreover, as fins are composed of live tissue, fin erosion can be classified as injury or damage to live tissue [51]. Fish possess nociceptors for the perception of painful stimuli in their fins, and fin damage may, therefore, be associated with pain [29,52]. Our results showed an important biological significance since fin condition relates directly with an injury to living tissue, and therefore a decrease in the welfare of fish. However, additional studies including indicators of pain should be done to estimate the full impact of *C. rogercresseyi* infestations on tissue damage. The potential use of the condition of dorsal, pectoral, and pelvic fins to monitor fish welfare during *C. rogercresseyi* infestations, could inform management strategies with the least impact on fish and avoiding detrimental effects on welfare. This indicator could be of great use in farms and laboratory facilities. On farms, fin condition could provide some information on well-being, although the cause may correspond to more than *C. rogercresseyi* infestations, as other factors that can also induce fin deterioration. However, under laboratory conditions, this indicator can become an important tool for the welfare monitoring of laboratory fish, particularly when extended maintenance of the parasite is required.

## 5. Conclusions

The association of fin erosion, increasing fish stress and parasite abundances was presented in this study, suggesting that *C. rogercresseyi* infestation is one of the possible causes for fin degeneration observed in Atlantic salmon. Pectoral and pelvic fins were the most affected and damaged and were related to increasing stress (glucose levels, also the dorsal fin). Further, our results suggest that pectoral, pelvic, and dorsal fins may be used as good indicators of fish welfare related to parasitosis. The evaluation of fin condition is a reliable, noninvasive, repeatable, and low-cost method for measuring fish welfare during *C. rogercresseyi* infestation. The knowledge obtained during this study may be useful for increased welfare in farmed fish, but mostly for fish cultivated under laboratory conditions for experimental infestations with *C. rogercresseyi*.

## Figures and Tables

**Figure 1 animals-10-01166-f001:**
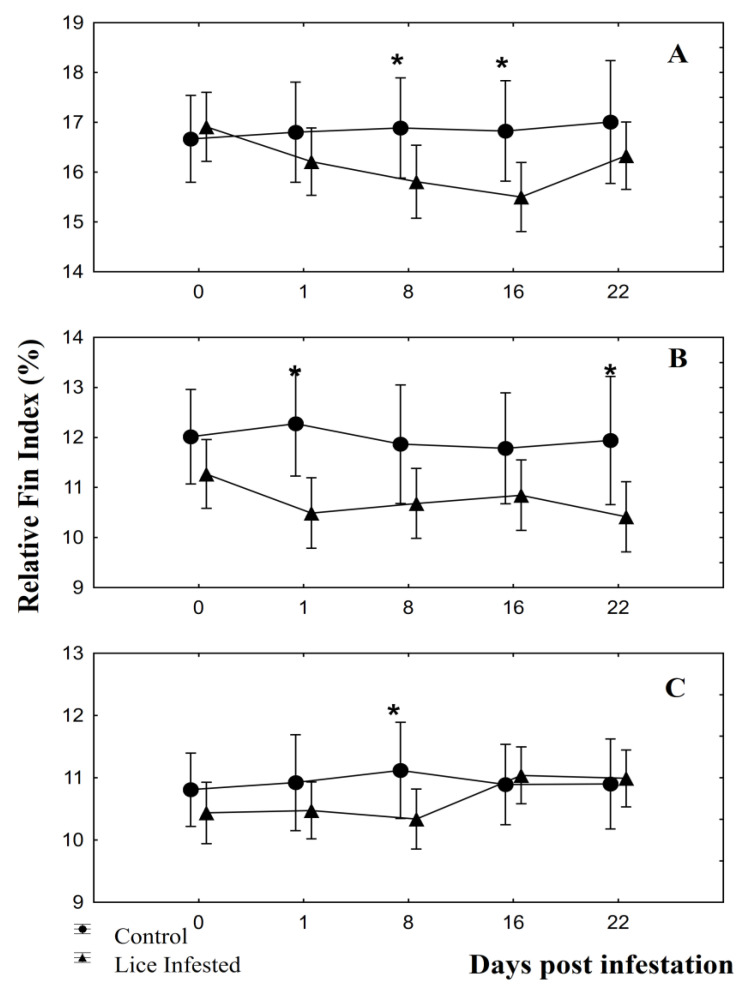
Relative fin index (%) of control and *Caligus rogercresseyi* infested groups of Atlantic salmon (*Salmo salar*), corresponding to (**A**) pectoral fins, (**B**) pelvic fins, and (**C**) anal fin during the time curse (days post infestation). Mean ± 0.95 Confidence Interval. Asterisks show significant differences (*p* < 0.05) compared to the fish in the control group on that day.

**Figure 2 animals-10-01166-f002:**
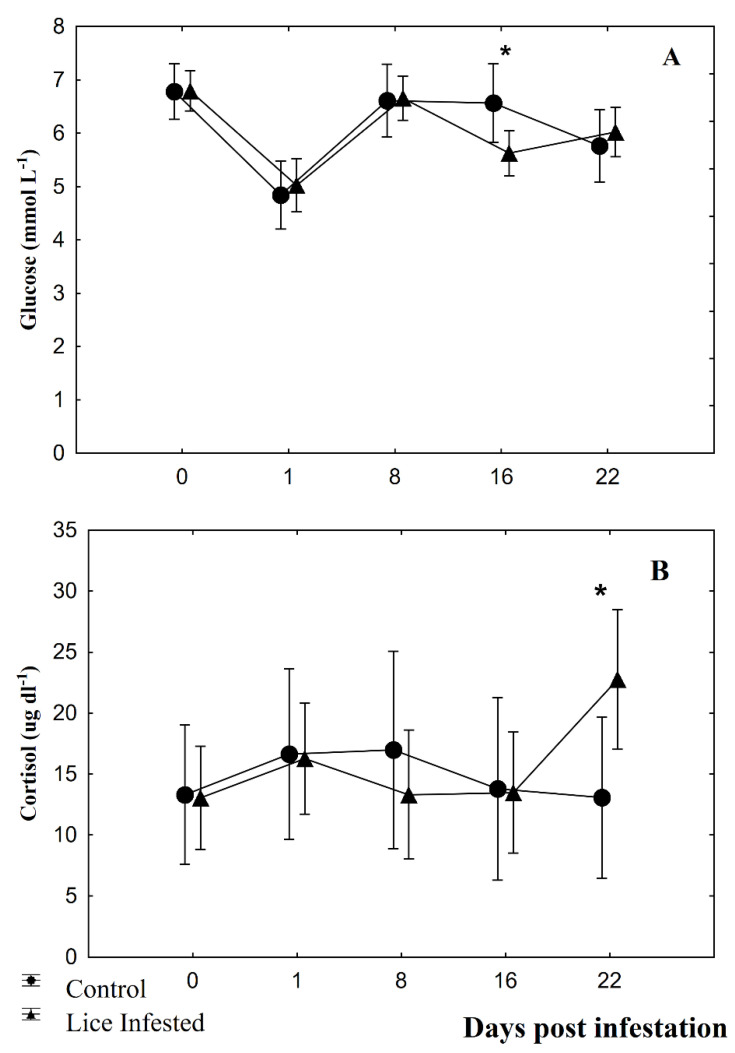
Physiological plasmatic parameters of control and *Caligus rogercresseyi* infested groups of Atlantic salmon (*Salmo salar*), corresponding to (**A**) glucose, and (**B**) cortisol levels during the time curse (days post infestation). Mean ± 0.95 Confidence Interval. Asterisks show significant differences (*p* < 0.05) compared to the fish in the control group on that day.

**Figure 3 animals-10-01166-f003:**
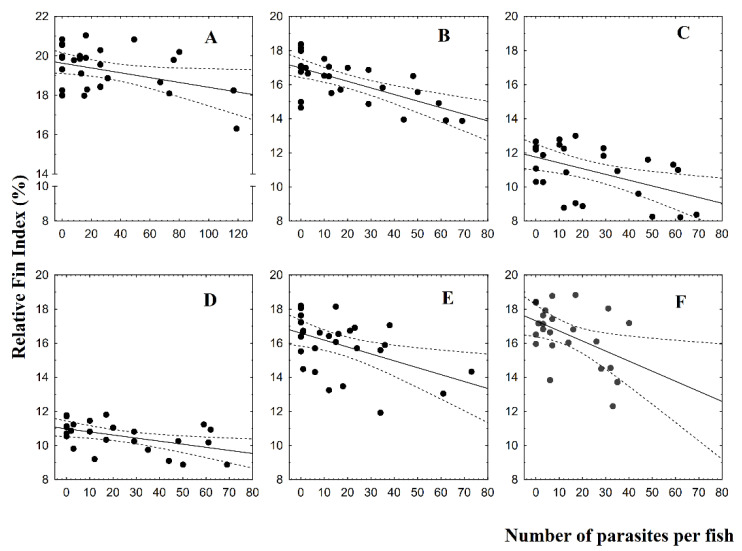
Linear relationships between the mean relative fin index (%) and *Caligus rogercresseyi* abundances of Atlantic salmon (*Salmo salar*) for (**A**) caudal fin at 1 dpi, (**B**) pectoral, (**C**) pelvic, (**D**) anal fin at 8 dpi, and pectoral fin (**E**) at 16 dpi, and (**F**) at 22 dpi.

**Figure 4 animals-10-01166-f004:**
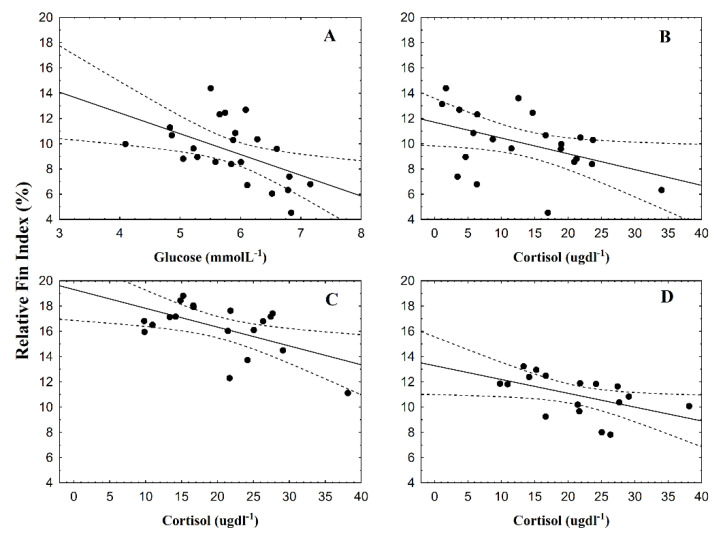
Linear relationships between the mean relative fin index (%) and *Caligus rogercresseyi* plasma glucose and cortisol levels of Atlantic salmon (*Salmo salar*) for (**A**,**B**) dorsal fin at 16 dpi, (**C**) for pectoral at 22 dpi, and (**D**) pelvic fin at 22 dpi.

**Table 1 animals-10-01166-t001:** Summary of the number and frequency (%) of each developmental stage of *Caligus rogercresseyi* in all infested fish throughout the experiment.

dpi	Cop	Ch I–II	Ch III–IV	AF	AM
1	825 (100)	0	0	0	0
8	0	596 (100)	0	0	0
16	0	0	482 (97)	10 (1)	20 (2)
22	0	0	0	184 (46)	218 (54)

dpi: days postinfestation; Cop: copepodid; Ch I–II: first and second chalimus stages combined; Ch III–IV: third and fourth chalimus stages combined; AF: adult female; AM: adult male. % in brackets.

**Table 2 animals-10-01166-t002:** Mean *Caligus rogercresseyi* abundance per tank on sampled Atlantic salmon (*Salmo salar*) throughout the experiment.

dpi	Tank 1	Tank 2	Tank 3	Tank 4	Tank Mean
1	15.6 (± 6.73)	29.8 (± 22.35)	32.0 (± 25.10)	87.6 (± 30.05)	41.3 (±31.74)
8	8.2 (± 4.87)	18.4 (± 5.59)	32.0 (± 15.02)	60.2 (± 6.83)	29.8 (±22.55)
16	8.4 (± 5.41)	14.6 (± 6.54)	34.4 (± 17.10)	44.6 (± 20.08)	25.5 (±16.88)
22	2.6 (± 2.61)	10 (± 4.64)	22.2 (± 10.28)	44 (± 21.97)	19.8 (±18.10)

dpi: days postinfestation. Standard deviations are shown in parentheses. n = 5 fishes on each sampling occasion.

**Table 3 animals-10-01166-t003:** Mean relative fin index (%) of Atlantic salmon (*Salmo salar*) fin for both control and infested fish groups previous to the experiment (± standard deviation).

Relative Fin Index (%)	Control	Lice Infested
Dorsal fin	7.9 (± 3.5)	8.1 (±2.3)
Mean Caudal fins	19.9 (±1.1)	19.5 (±0.9)
Mean Pectoral fins	16.7 (±2.2)	17.0 (±1.9)
Mean pelvic fins	11.7 (±1.3)	10.9 (±1.8)
Anal fin	10.8 (±1.1)	10.1 (±2.1)

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
