# Peer review of "Fin Erosion of Salmo salar (Linnaeus 1758) Infested with the Parasite Caligus rogercresseyi (Boxshall & Bravo 2000)"

_animals, 2020, doi:10.3390/ani10071166_

Round 1
Reviewer 1 Report
It is an interesting manuscript reporting well planned and executed experiments.
The writing is somewhat verbose and in some instances confusing.
I made a plethora of comments on the manuscript, these are for clarity (word choice) and also to reduce repetitive sections and overstating of obvious matters. The hypothetical round circle type of comments have been deleted.

Author Response
Reviewer 1:
It is an interesting manuscript reporting well planned and executed experiments. The writing is somewhat verbose and in some instances confusing. I made a plethora of comments on the manuscript, these are for clarity (word choice) and also to reduce repetitive sections and overstating of obvious matters. The hypothetical round circle type of comments have been deleted.
Response: We appreciate the reviewer´s opinion and the many corrections that they made suggested. We think that the document was greatly improved thanks to their help.
Lines 25-28, 30, 31, 33-37: Changes were made according to the reviewer's suggestions.
Line 29: We understand that the correlation analysis looks for associations between variables. For this reason, we decided to keep the word.
Lines 41 – 43: The reviewer mentioned “this is a long stretch (remember correlation is not causation)”, we understand and agree with this statement. For this reason, we decided to change the lines to improve understanding.
The reviewer made several suggestions and corrections, so we decided to make the changes that they suggested. Although there are some points that we decided to keep or explain the changes we made.
Introduction section.
Line 54: the reviewer suggested the word “indigenous”, but this word means the same as native, so we decided to keep the word native in the text.
Lines 67, 68: the reviewer indicated that the citation was incomplete. We revised and agree with them. We made the necessary changes.
Lines 77-79: we tried to complete the sentence and we hope that now the mistake was corrected.
Methodology section.
Line 152: the reviewer asked about the samples. We added to which each sample corresponded to in the corresponding lines (due to changes in the number it is not the same).
Line 161: The reviewer asked if nets were also inspected, the answer is yes, it is a routine procedure, and also each tank has a unique net, and each time that the net was used, this was with clean seawater.
Statistical section: The reviewer asked about the word or statistical term “relationship” indicating cousins/aunts?? Or Correlation?.
In the study, we used simple linear regression analysis. According to Quinn & Keough 2002, this statistical model assumes a linear relationship between a continuous response variable and a single, usually continuous, predictor variable. Such models are termed simple linear regression models and their analysis has three major purposes: 1. to describe the linear relationship between Y and X, 2. to determine how much of the variation (uncertainty) in Y can be explained by the linear relationship with X and how much of this variation remains unexplained, and 3. to predict new values of Y from new values of X. Our experience is that biologists, especially ecologists, mainly use linear regression analysis to describe the relationship between Y and X and to explain variability in Y. The parameter called the correlation coefficient measures the strength of the relationship between the two variables.
We give this explanation to clarify why we used the term relationship, and we think that it is the appropriate analysis for this context. We hope that the reviewer -now with this explanation- will agree with us.
Reference: Quinn, G.P.; Keough, M.J. Experimental Design and Data Analysis for Biologists; Cambridge University Press: United Kingdom, 2002; pp. 537.
Results section:
The reviewer asked: What is r2 in all the above?. R-square is a statistical measure of how close the data is to the fitted regression line. In general, the higher the R-square, the better the model fits the data. However, in our study, this parameter was below 50% for all the estimated regression estimated. But, we considered that this result was not that bad. Why?. There are two main reasons why low R-square values ​​could be considered adequate. In some fields, the R-squared values ​​are expected to be low. For example, any discipline that attempts to predict human behavior, such as psychology, typically has R-square values ​​of less than 50%. Human beings are simply more difficult to predict than, for example, physical processes. Furthermore, if the R-square value is low but they predictors are statistically significant (which is the case, we have other parameters with good results), you can still draw important conclusions about the association between changes in predictor values ​​and changes in the response value. Regardless of the R-square, the significant coefficients still represent the mean change in response to one unit of change in the predictor, while the other predictors in the model are held constant. This type of information can be very valuable (Minitab, 2019).
This parameter has a statistical nature, being the reason why we did not want to write much about it.
https://blog.minitab.com/es/analisis-de-regresion-como-puedo-interpretar-el-r-cuadrado-y-evaluar-la-bondad-de-ajuste

Reviewer 2 Report
This paper describes the effect of the lice parasite, Caligus rogercresseyi on the fins of Salmo salar and as such it makes a significant contribution to our knowledge on the damage caused by this important pathogen. The report is well written and presents a set of interesting observations. The methodology is good and there is satisfactory critical analysis of the data.
I have only one comment which needs attention by the authors. Figures 1 and 2, like the majority of the data, show considerable variation with, at times, large 95% confidence limits. This occurs even at points which show significance e.g. Fig 1. The authors are to be complimented on their extensive description of the statistical methods used, but I note in Fig 1 Relative Fin index is based on low percentages with a considerable range of 95% confidence limits. Significance in this data set forms the basis of the hypothesis presented and is related to the damage caused by the parasite on the fins. I was wondering, given the low percentages, whether the data presented ought to be transformed prior to statistical analysis e.g. arcsin?
Author Response
Reviewer 2:
This paper describes the effect of the lice parasite, Caligus rogercresseyi on the fins of Salmo salar and as such it makes a significant contribution to our knowledge on the damage caused by this important pathogen. The report is well written and presents a set of interesting observations. The methodology is good and there is satisfactory critical analysis of the data.
Response: We appreciate the reviewer´s comments. We are grateful to know that there is another researcher that finds our study relevant.
I have only one comment which needs attention by the authors. Figures 1 and 2, like the majority of the data, show considerable variation with, at times, large 95% confidence limits. This occurs even at points which show significance e.g. Fig 1. The authors are to be complimented on their extensive description of the statistical methods used, but I note in Fig 1 Relative Fin index is based on low percentages with a considerable range of 95% confidence limits. Significance in this data set forms the basis of the hypothesis presented and is related to the damage caused by the parasite on the fins. I was wondering, given the low percentages, whether the data presented ought to be transformed prior to statistical analysis e.g. arcsin?
Response: We appreciate the reviewer´s suggestion, however, we don’t think that is necessary to transform data for the stabilization of the variance, as a consequence, yes, our CI of 95% is wide, but we don’t think that this is something to worry about. The following paragraph is our argument:
According to Wheeler et al., 2006, some researchers might compare confidence intervals (CIs) from two populations, and if these CIs overlap, they conclude that the populations are not statistically different. The use of the CI overlap test is driven by the simplicity and availability of CIs as part of standard statistical software, and this test is used in situations when the software does not perform the appropriate tests on the statistics of interest. This is a flawed decision rule that results in an overly conservative testing strategy. Schenker and Gentleman [1] provided a comprehensive analysis of the statistical inadequacy of using a CI overlap testing procedure. In particular, they concluded that ‘‘[a CI overlap testing procedure] should not be used for formal significance testing unless the data analyst is aware of its deficiencies and unless the information needed to carry out a more appropriate procedure is unavailable’’ [1, p 186]. Although the CI overlap method is flawed, making it more difficult to detect significant differences, it is used frequently and reported throughout the literature.
Reference: Wheeler, M.W.; Park, R.M.; Bailer, A.J. Comparing median lethal concentration values using confidence interval overlap or ratio tests. Environmental Toxicology and Chemistry: An International Journal 2006, 25, 1441-1444.
For these reasons, we decided to keep the data and analysis as it was presented in the first place and we trust the statistical differences observed despite the wide CI95.

Round 2
Reviewer 1 Report
I checked the response letter and the ms and am satisfied with the authors' response. I attach an annotated letter. My comments are in blue.
Regards
